# Analysis of Surface State after Turning of High Tempered Bearing Steel

**DOI:** 10.3390/ma15051718

**Published:** 2022-02-24

**Authors:** Mária Čilliková, Anna Mičietová, Róbert Čep, Martina Jacková, Peter Minárik, Miroslav Neslušan, Karel Kouřil

**Affiliations:** 1Faculty of Mechanical Engineering, University of Žilina, Univerzitná 1, 01026 Žilina, Slovakia; anna.micietova@fstroj.uniza.sk (A.M.); miroslav.neslusan@fstroj.uniza.sk (M.N.); 2Faculty of Mechanical Engineering, VŠB—Technical University of Ostrava, 17. Listopadu 2172/15, 70800 Ostrava, Czech Republic; robert.cep@vsb.cz; 3Research Centre, University of Žilina, Univerzitná 1, 01026 Žilina, Slovakia; martina.jackova@uniza.sk; 4Faculty of Mathematics and Physics, Charles University, Ke Karlovu 5, 12116 Prague, Czech Republic; peter.minarik@mff.cuni.cz; 5Faculty of Mechanical Engineering, Brno University of Technology, Technická 2896/2, 61669 Brno, Czech Republic; karel.kouril@vutbr.cz

**Keywords:** white layer, microhardness, turning, surface state

## Abstract

This paper investigates surface state after turning of the high tempered bearing steel 100Cr6 with a hardness of 40 HRC. White layer (WL) thickness and its microhardness, as well as surface roughness, are investigated as a function of tool flank wear *VB* as well as cutting speed *v_c_*. The mechanical and thermal load of the machined surface were analysed in order to provide a deeper insight into their superimposing contribution. Cutting energy expressed in terms of cutting force was analyses as that consumed for chip formation *F**_γ_* and consumed in the flank wear land *F**_α_*. It was found that the mechanical energy expressed in terms of the shear components of the *F**_α_* grows with *VB*, converts to the heat and strongly affects the thickness of the re-hardened layer. Furthermore, the superimposing contribution of the heat generation and its duration in the *VB* region should also be taken into account. It was also found that the influence of *VB* predominates over the variable cutting speed.

## 1. Introduction

Increasing demands on components’ functionality has led to novel concepts for their manufacturing, together with the application of progressive materials [1,2,3]. The high tempered steels undergo the conventional quenching process when a component is rapidly cooled down from a high temperature, followed by tempering in the furnace at an elevated temperature. The final hardness and the corresponding microstructure is usually a function of tempering temperature. Such heat treatment produces a matrix in which the outstanding combination of hardness and toughness is mixed. Components made of these steels are produced mostly via machining cycles and a subsequent final surface treatment. The term hard machining (hard turning) is usually associated with the steels of hardness above 50 HRC. On the other hand, some aspects predominant in hard turning cycles can be found during machining of the steels below this threshold as well, especially the formation of a surface white layer (WL) as a re-hardened matrix [4]. WL is produced on machined surface when the temperature in the flank wear land is above 770 °C followed by the high cooling rate. Such conditions are met when flank wear *VB* exceeds a certain threshold [2,4]. Thermally softened layer (dark region) lying below the near surface WL usually cannot be recognized in the case of high tempered samples due to the shadowing effect of the previous tempering in the furnace during heat treatment. WL thickness is affected mainly by the size of *VB* region and the main cutting motion *v_c_* [4,5,6,7,8]. In contrast with grinding, dislocation density in WL after turning is usually higher, causing severely strained matrix [4,8,9]. Despite high temperatures and superimposing high stresses, the size of the carbides inside WL remains unchanged, whereas martensite matrix is refined down to several tens of nanometers [8,10,11,12]. Li et al. [11] also reported the steep grain size gradient in the near surface region. Ramesh et al. [12] investigated microstructure of WL of hardened steel 52100 and discussed the significance of thermal cycle. The deep insight into thermal cycle during formation of WL was reported by Hosseini et al. [13] as well. It should be mentioned that the surface state of quenched steel remarkably affects its functional properties as was reported in new studies [14,15,16]. Contribution of WL depends on the manner in which surface after hard turning is exploited [17,18]. Presence of WL is acceptable in the sliding contact whereas the cycling rolling contact is quite sensitive to the surface re-hardening [17,18,19].

Machining cycles are usually performed at constant cutting conditions in production of real industrial components whereas tool wear and the corresponding cutting edge geometry change. Developed deviations in tool geometry affect energy consumption and the corresponding cutting force. It should be noted that this energy is converted into the heat and high temperatures in the cutting zone strongly affect the surface state of turned parts. It was reported that WL at low *VB* is low or missing [20], followed by continuous increase in cutting force and WL thickness at higher *VB*. However, measured components *F_c_* and *F_p_* are composed of those consumed in the tool-chip interface (*F_γ_*) and flank wear land (*F_α_*). For this reason, only the shear and normal components of *F_α_* should be involved in correlation analyses in which surface state of produced parts is analysed.

Wang and Liu [8] employed Green’s function in order to distinguish between *F_α_* and *F_γ_* and their normal and shear components). The authors demonstrated the strong correlation between *VB* and WL thickness as well as the growth of *F_α_* and moderate decrease in *F_γ_* when flank wear becomes more developed. However, Neslušan et al. [7] reported on the methodology for experimental decomposition of cutting force components and indicated that the components of *F_γ_* grow due to crater wear. The authors carried out the measurement of cutting force components and their decomposition as a function of *VB.* It is worth mentioning that cutting force components (especially their shear components) can be linked with the thermal load of the machined surface as almost the whole mechanical energy consumed by the turning process is converted into heat. For this reason, cutting force decomposition also provides information about the superimposing thermal effects.

This study investigates the surface state of the high tempered bearing steel after turning. The investigation is focused on the observation of surface roughness, WL thickness and its microhardness as a function of *VB*, as well as cutting speed *v_c_* based on the measurement of cutting force components and their decomposition, as reported in [7]. Finally, the thermal load of the produced surfaces is discussed as well. As contrasted with the previous studies [5,6,8], this study provides deeper insight into pure contribution of the shear and normal components in the different regions and their relationship to the thermal load. As compared with the study of Wang and Liu [8] or our previous work [7], this study also discusses contribution of the thermal cycle duration, as well as the role of specific heat to the thickness of WL. Finally, as compared with [7], thermal cycle and the produced WL are investigated as a function of cutting speed and supplemented by the measurement of microhardness in WL.

## 2. Experimental Setup

The experiments were performed on the high tempered steel 100Cr6 (40 ± 2 HRC, temperature of austenitisation 840 °C, quenching temperature 60 °C followed by 2 h tempering at 530 °C). The sample dimensions were: outer diameter 55 mm, inner diameter 40 mm, and length 70 mm. Details associated with turning process are listed in Table 1. Cutting feed, depth, and speed were chosen on the base of recommendations provided by the producer of inserts. Moreover, these cutting conditions can be considered as the conventional parameters for hard machining cycles, especially when the cutting depth and feed should be kept low with respect of thermal softening effect in the cutting zone.

The inserts with sufficiently high *VB* were used in this study to make WL as thick as possible. Furthermore, the influence of variable cutting speed (as the additional aspects remarkably contributing to WL thickness) was investigated as well.

The decomposition of cutting force components is illustrated in Figure 1. Separation of the shear and normal components *F_αt_*, *F_αtn_*, *F_γt_*, and *F_γtn_* was carried out by the use of a sample with a variable cutting depth [6]. Further details about the methodology, the employed equations, a brief sketch of the turned sample, and the appearance of tool wear land, can be found in the previous study [7].

The cutting edge geometry alterations were investigated by Alicona 5 device (IF-Edge MasterModule) (Alicona Imaging GmbH, Graz, Austria) at the different phases of *VB*. The rake angle *γ**_n_* as well as the cutting edge radius *r_n_* were obtained from 25 measurements regularly distributed along the cutting edge length. Chip’s morphology was observed in scanning electron microscope (SEM) ZEISS Auriga Compact (AZoNetwork UK Ltd., Manchester, UK). The surface roughness was measured by Hommeltester T 2000 (Hommelwerke GmbH, Schwenningen, Germany) along 4 × 0.8 mm length (probe T 100).

Measurement of *F_c_* as well as *F_p_* was carried out by the use of a dynamometer Kistler 9441 (sampling frequency 2 kHz, DasyLab software- measX GmbH, Aachen, Germany). *F_c_* as well as *F_p_* were obtained by averaging the three repetitive passes after the signal filtration (the low pass filter 20 Hz). To reveal the microstructural transformations induced by hard turning (for *a_p_* = 0.25 mm and direction of *F_c_*), 10 mm long pieces were prepared for metallographic observations (etched by 5% Nital for 8 s). Microhardness measurement of HV 0.05 in WL was carried out only for samples turned by the insert of *VB* = 0.8 mm because the thickness of WL was sufficiently high only for this *VB*. Microhardness measurements were carried out using an Innova Test 400 TM (50 g for 10 s) (Innovatest, Maastricht, The Netherlands) on the cross sectional cuts after metallographic observations. Five repetitive measurements of WL thickness, microhardness, and surface roughness were carried out (average values as well as the corresponding standard deviations were calculated and provided in the paper).

## 3. Results and Discussion

The recorded cutting force evolutions (as that illustrated in Figure 2) show five different stages. The initial stage is associated with zero cutting depth (elastic deformation of the produced surface only [21]) whereas the consecutive stages represent the increasing *a_p_* in which chip is generated. Measured *F_c_* as well as *F_p_* are progressively growing along with *a_p_* but saturation of this growth can be observed at higher *VB* and *a_p_*.

The evolution of *VB* versus a*_p_* appears to be linear (see Figure 3). It is known that the cutting forces grow along with cutting depth is driven by the power law but the exponent for the cutting depth is near by 1 [3]. For this reason, the linear fit for the measured components is reasonable. The increase in *F_p_* is higher than *F_c_*. A much steeper increase can be found for *VB* = 0.8 mm, whereas only a moderate one can be reported for lower *VB* (especially in the case of *F_c_*). The similar evolution can be found for the extracted components *F*_*αt*_ and *F*_*αtn*_ (see the values in Table 2). The average friction coefficient (*F_αt_*/*F_αtn_*) drops down along with *VB* due to predominate grow of *F_αtn_* components (see also the values in Table 2). Note that the values in Table 2 represent the average values in the flank wear land. In reality, a heterogeneous distribution is observed [22] but the simplified uniform model is employed in this particular case.

It was reported [8] that *F_γ_* moderately decreases along with *VB* in hard tuning. On the other hand, it was also demonstrated [23,24] that the components associated with chip formation are unchanged when turning annealed steel. It should be considered that the process of tool wear is complex and involves the alterations of tool rake geometry as well (see appearance of the cutting edge in [7]). For this reason, *F_γ_*_c_ and *F_γp_* also grow (see Figure 4) as a results of altered rake angle and cutting edge radius (see Table 2). Figure 4 also depicts that the changes in the rake angle geometry strongly affect the normal components of *F_γ_* as compared with the shear one.

Plasticity in the cutting zone in the case of hardened matrix is due to its self-heating (matrix softens and becomes formable [25,26]) as well as the superimposing high hydrostatic pressure [27]. However, matrix softening occurs near the cutting edge only and vanishes along with increasing distance from the tool-chip interface. For this reason, *F_γc_* and *F_γp_* are growing versus *a_p_*. The changes in shear instability and the corresponding shape of produced segmented chips (see Figure 5) are due to the alterations in thermal and stress state in the tool-chip interface as a result of altered tool rake geometry [2,7,28]. It can be seen that the more developed crater wear increases the distance between the neighbouring segments, their size, and degree of chip segmentation [7], as well as the corresponding segmentation frequency [3]. This is a result of prolonged accumulation of stress ahead of the cutting edge and its abrupt release when the segment is initiated by the brittle cracking on the free surface [25]. Such behaviour has been already reported [3,28].

Figure 6 shows the typical WL produced during turning of hardened as well as tempered steels. Its white colour is due to flatness of the surface after etching and the corresponding missing phases contrast. Presence of the re-hardened layer indicates that the machined surface temperature exceeds austenitising temperature followed by the rapid cooling [2,14]. WL for lower *VB* is discontinuous and quite thin, progressively growing with *VB*, see Figure 6. Figure 7 also shows that a local minimum occurs in the region of insert wear in which the tool geometry is markedly altered, see also Figure 6. Figure 7 also depicts that WL thickness grows along with increasing *v_c_* as well, but the contribution of *VB* prevails.

The growth of WL thickness fits well with findings of Chaudhari and Hashimoto [20]. Authors reported that growth of the normal force is only sluggish in the initial phase of tool wear in which WL cannot be found, followed by a much steeper increase in force as soon as the continuous WL increases in thickness along with increasing *VB*. Figure 7 and Table 2 indicate the low *F_αt_* and *F_αtn_* corresponds with the low thickness of WL and vice versa. Moreover, the altered cutting edge shape and geometry, together with the increasing cutting force components, affect the new surface generation expressed in terms of its topography (surface roughness), see Figure 8 and Figure 9. Increasing intensity of machined material side flow increases the height of surface irregularities and the corresponding values of *Ra*. The values of the shear and normal components per 1 mm of *VB* (last two columns in Table 2) indicate that the straightforward correlation between the high *F_αt_* and *F_αtn_* and WL thickness is quite debatable as the specific mechanical load per 1 mm of *VB* decreases along with *VB*. Expressed in other words, it can be reported that the average stress in the *VB* region decreases with *VB*. For this reason, the thermal effect and its duration should also be taken into account.

The time interval *τ* during which the produced surface is exposed to the elevated temperatures is driven by *v_c_* and *VB* as follows:(1)$$\tau \;=\frac{VB}{v_{c}}\;\left(\mathrm{ms}\right)$$

The heat generated in the *VB* region *Q*_α_ can be calculated as follows:(2)$$Q_{\alpha }=F_{\alpha t}\cdot v_{c}\;({\mathrm{kJ}\cdot \min}^{-1})$$

Calculations and Figure 10 show that WL thickness only increases with *τ* when the *v_c_* is kept constant. As soon as the influence of *v_c_* is considered, the pure contribution of *τ* is quite controversial. The increasing *v_c_* reduces the time *τ* during which the produced surface is exposed to the heat (and the corresponding elevated temperatures) but WL thickness increases. Therefore, the superimposing contribution of *τ* and *Q_α_* (see Figure 11) should be taken into account and specific heat *Q_α_*′ should be calculated as follows:(3)$${Q_{\alpha }}^{\prime }=F_{\alpha t}\cdot v_{c}\cdot \tau \;({\mathrm{kJ}\cdot \min}^{-1}\cdot \mathrm{ms})$$

Figure 12 depicts that the specific heat *Q_α_′* better explains the contribution of the thermal load to the WL thickness. WL thickness increases with *v_c_* progressively for all *VB*. However, the evolution of *VB* versus *Q_α_*′ with respect of WL thickness shows that a lower WL thickness is obtained for *VB* = 0.6 mm despite the higher *Q_α_*′ as compared with *VB* = 0.4 mm. Such an evolution strongly correlates with the evolution of the *F_αtn_* components (see Figure 4) which indicates that the formation of WL is not a pure product of the thermal cycle (austenitising followed by rapid self-cooling), but that the influence and superimposing contribution of the mechanical load should be considered as well. Expressed in other words, the phase transformation in the *VB* region is under stress which affects its dynamics, extent, and hardness. Figure 13 demonstrates that the microhardness HV 0.05 in WL progressively drops down with *v_c_*. The thicker the WL is produced, the lower the microhardness is and vice versa, see Figure 14. It is considered that the microhardness of WL is mainly driven by (i) the rate of cooling which should be higher at lower *v_c_* due to the lower accumulation of thermal energy expressed in *Q_α_*′ and (ii) the superimposing contribution of higher *F_αtn_*.

The thermal softening which also affects the deeper regions below the near surface WL is only minor (in this particular case) as this effect is shadowed by the previous thermal softening during samples tempering in the furnace.

## 4. Conclusions

It can be concluded that the influence of cutting edge wear on hard turning is complex. The crater wear alters the shear instability in the tool-chip interface and the corresponding chip appearance. On the other hand, *F_αt_* and *F_αtn_* grow with VB and contribute to the thicker WL. However, WL is a product of energy consumed in the tool flank region and the superimposing thermal cycles. The time within the machined surface is exposed to heating plays significant role together with the specific heat strongly affected by cutting speed.

From the practice point of view, it can be reported that VB should not exceed 0.2 mm. The WL in this region is discontinuous or its thickness is low. Employment of the insert of higher VB results into remarkable increase in WL thickness. Employment of the inserts of higher VB could be allowed for roughing cycles only as the thick WL will be removed during the consecutive grinding or turning finishing. Intensification of hard turning cycles via higher cutting speed increases WL thickness but drops down its microhardness.

## Figures and Tables

**Figure 1 materials-15-01718-f001:**
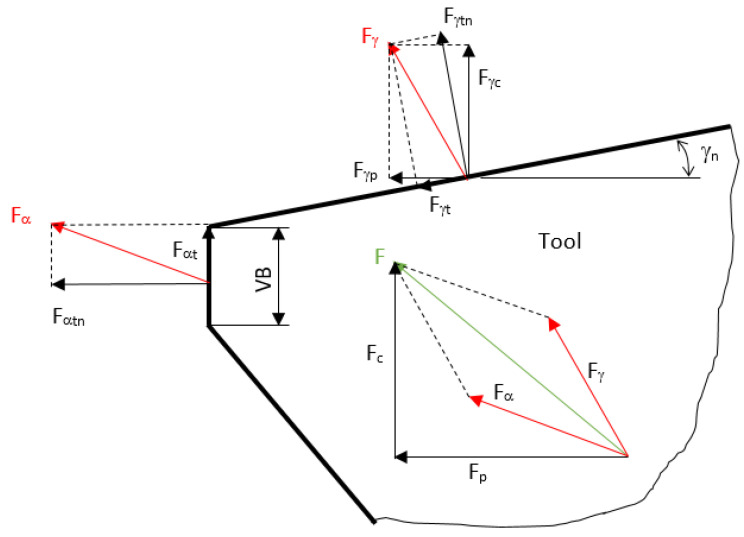
Decomposition of cutting force in the cutting zone.

**Figure 2 materials-15-01718-f002:**
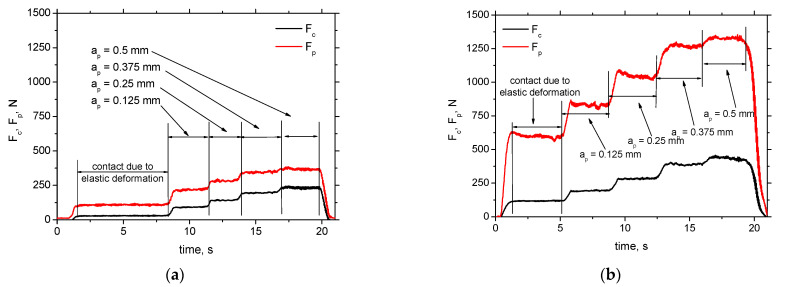
Record of cutting force components *F_c_* and *F_p_* as a function of cutting depth (low pass filter 20 Hz). (**a**) *VB* = 0.1 mm, (**b**) *VB* = 0.8 mm.

**Figure 3 materials-15-01718-f003:**
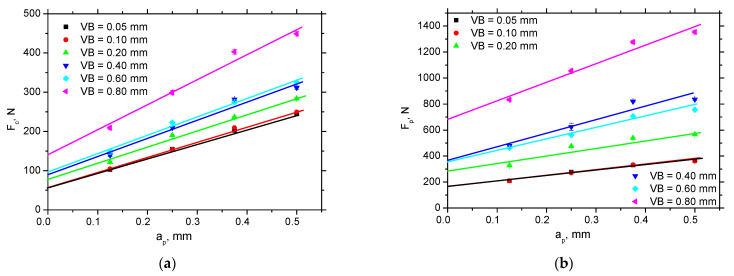
*F_c_* and *F_p_* versus a*_p_*. (**a**) *F_c_* versus *a_p_*, (**b**) *F_p_* versus *a_p_*.

**Figure 4 materials-15-01718-f004:**
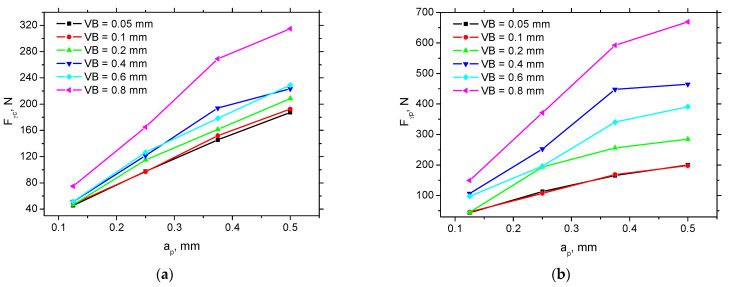
Evolution of *F*_γc_ and *F*_γp_, *a_p_* = 0.25 mm. (**a**) *F**_γ_**_c_* versus *a_p_*, (**b**) *F**_γ_**_p_* versus *a_p_*. Note: the standard deviations of obtained *F_γc_* is ± 7 N and ± 13 N for *F_γp_*.

**Figure 5 materials-15-01718-f005:**
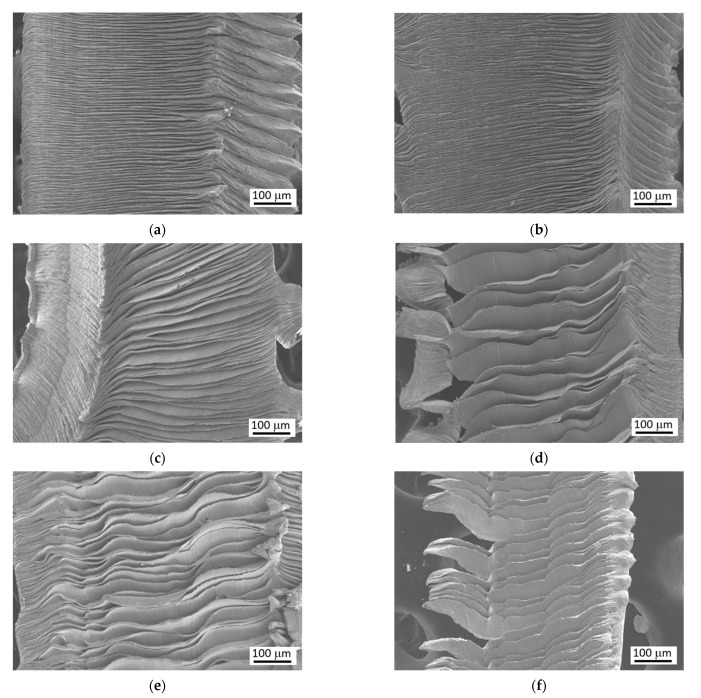
SEM observation of produced chips. (**a**) *VB* = 0.05 mm, *γ_n_* = −32°, *r_n_* = 47 μm, (**b**) *VB* = 0.1 mm, *γ_n_* = −26°, *r_n_* = 44 μm, (**c**) *VB* = 0.2 mm, *γ_n_* = −22°, *r_n_* = 35 μm, (**d**) *VB* = 0.4 mm, *γ_n_* = −2°, *r_n_* = 22 μm, (**e**) *VB* = 0.6 mm, *γ_n_* = −13°, *r_n_* = 67 μm, (**f**) *VB* = 0.8 mm, *γ_n_* = −12°, *r_n_* = 62 μm.

**Figure 6 materials-15-01718-f006:**
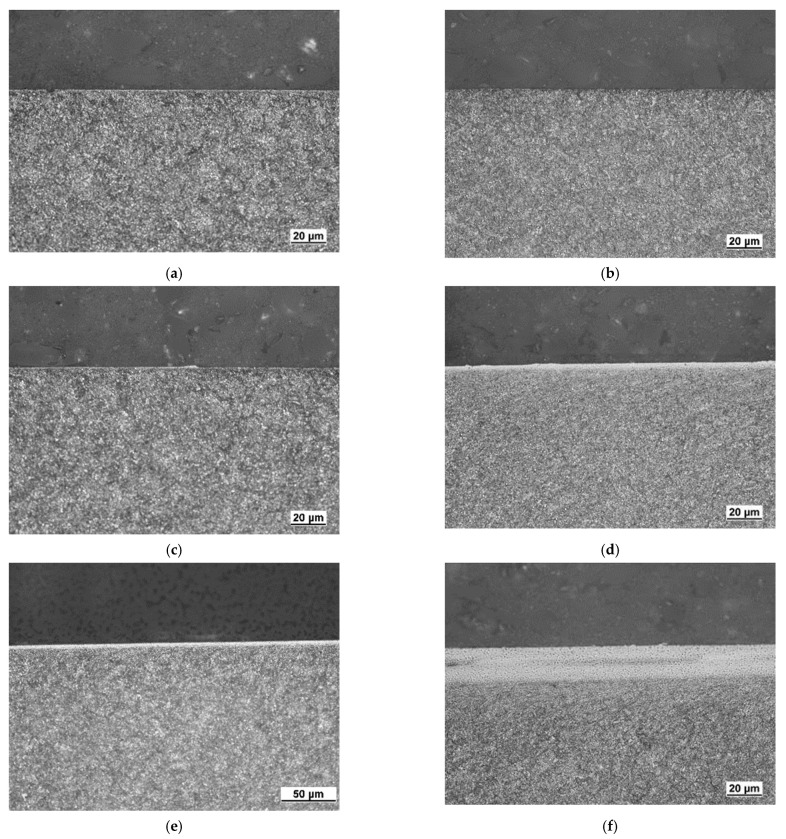
Metallographic images of machined surfaces, *v_c_* = 100 m.min^−1^. (**a**) *VB* = 0.05 mm, (**b**) *VB* = 0.1 mm, (**c**) *VB* = 0.2 mm, (**d**) *VB* = 0.4 mm, (**e**) *VB* = 0.6 mm, (**f**) *VB* = 0.8 mm.

**Figure 7 materials-15-01718-f007:**
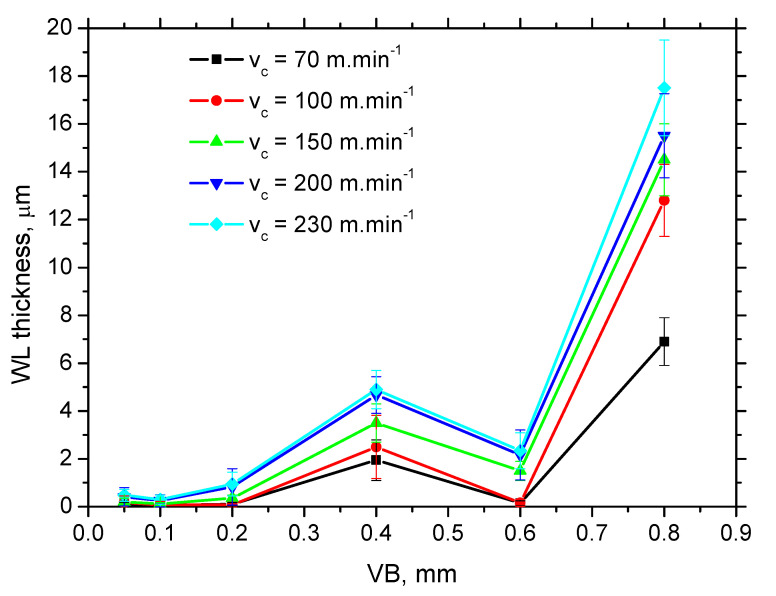
WL thickness as a function of *v_c_* and *VB*.

**Figure 8 materials-15-01718-f008:**
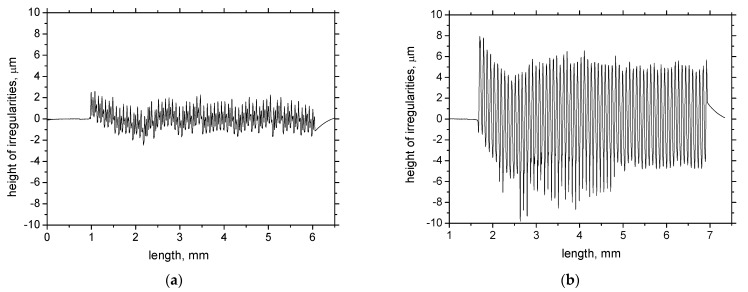
Records of height of irregularities for different *VB*. (**a**) *VB* = 0.05 mm, (**b**) *VB* = 0.6 mm.

**Figure 9 materials-15-01718-f009:**
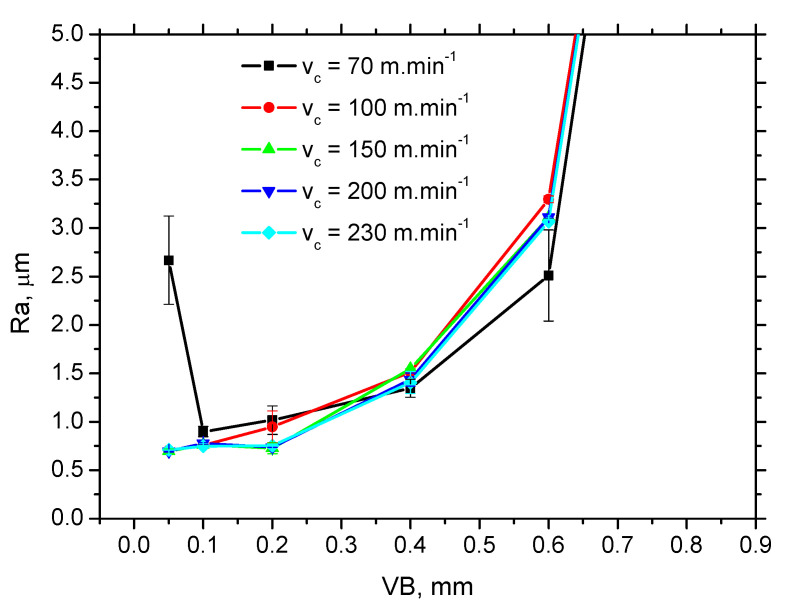
*Ra* as a function of *v_c_* and *VB*.

**Figure 10 materials-15-01718-f010:**
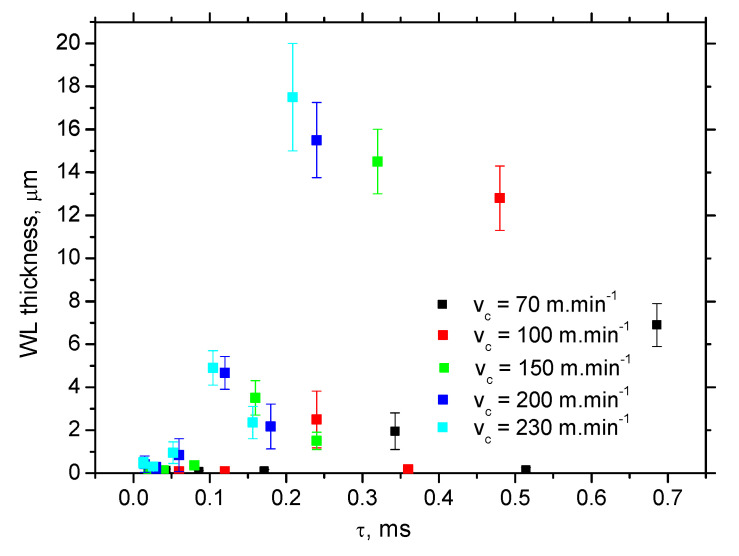
Time interval *τ* during which the produced surface is exposed to the elevated temperatures.

**Figure 11 materials-15-01718-f011:**
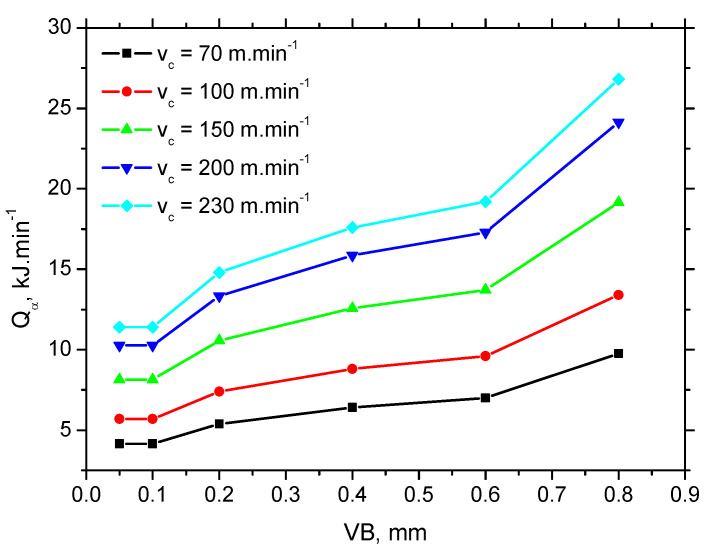
Heat *Q_α_* generated in the *VB* region.

**Figure 12 materials-15-01718-f012:**
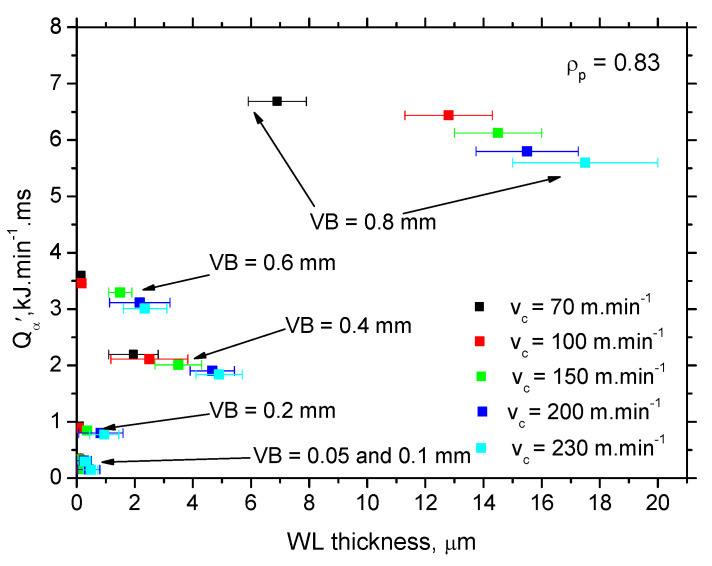
Specific heat *Q_α_*′ generated in the *VB* region.

**Figure 13 materials-15-01718-f013:**
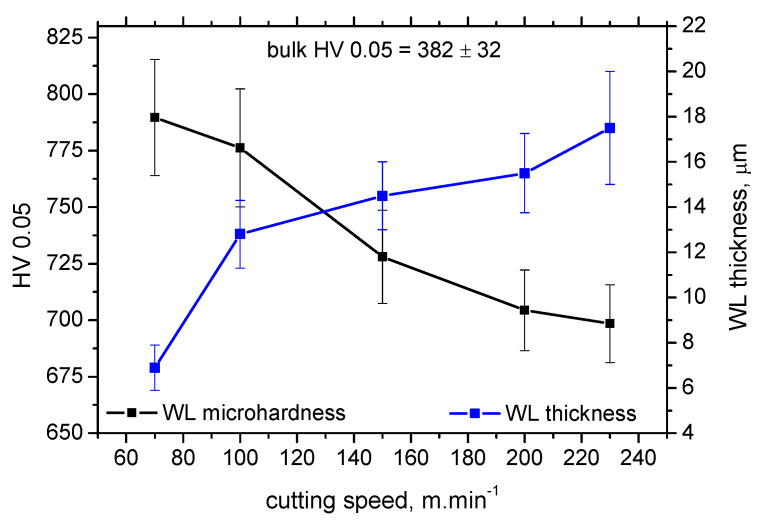
Microhardness and WL thickness as a function of *v_c_*, *VB* = 0.8 mm.

**Figure 14 materials-15-01718-f014:**
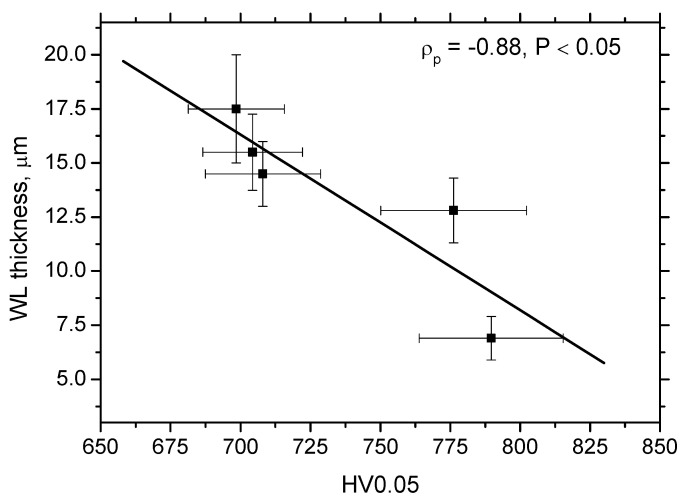
WL thickness versus microhardness HV0.05, *VB* = 0.8 mm.

**Table 1 materials-15-01718-t001:** Cutting and other conditions of turning process.

Cutting insert	DNGA 150,408, PCBN, 70% of CBN, TiN coated
Inset geometry	*r_ε_* = 0.8 mm, chamfer angle −35°, chamfer width 250 μm
Flank wear *VB*	from 0.05 to 0.8 mm (prepared before the test)
Feed *f*	0.09 mm
Cutting depth *a_p_*	from 0.125 to 0.5 mm
Cutting speed *v_c_*	from 70 to 230 m·min^−1^
Cutting environment	dry machining

**Table 2 materials-15-01718-t002:** Shear and normal components of *F**_α_*, rake angle *γ**_n_*, and cutting edge radius *r_n_* as well as *F**_α_**_t_* and *F**_α_**_tn_* per 1 mm of *VB*.

*VB*, mm	*F**_αt_*, N	*F**_αtn_*, N	*F**_αt_*/*F**_αtn_*	*γ_n_*, °	*r_n_*, μm	*F**_αt_*,/*VB*, N·mm^−1^	*F**_αtn_*,/*VB*, N·mm^−1^
0.05 ± 0.01	57	164	0.35	−32 ± 1.2	47 ± 8	1140	3280
0.1 ± 0.013	57	163	0.35	−26 ± 3.7	44 ± 7	570	1630
0.2 ± 0.025	74	282	0.26	−22 ± 4.0	35 ± 7	370	1410
0.4 ± 0.030	88	372	0.23	−2 ± 3.6	22 ± 7	220	930
0.6 ± 0.038	96	265	0.26	−13 ± 3.3	67 ± 12	160	608
0.8 ± 0.055	134	684	0.19	−12 ± 4.0	62 ± 13	168	855

Note: the standard deviations of obtained *F**_αt_* is about ± 7 N and ± 13 N for *F**_αtn_*.

## Data Availability

The raw data required to reproduce these findings cannot be shared easily due to technical limitations (some files are too large). However, authors can share the data on any individual request (please contact the corresponding author by the use of their mailing address).

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
