# Peer review of "Analysis of Surface State after Turning of High Tempered Bearing Steel"

_materials, 2022, doi:10.3390/ma15051718_

Round 1

Reviewer 1 Report

The authors investigated the state of the machined surface (White layer (WL) thickness and its microhardness as well as surface roughness) for high tempered bearing steel after the turning process. In addition, the thermal load of the produced surfaces has been pointed. The article is very interesting. However, the following points need to be addressed carefully for the improvement of the manuscript.

  1. Add literature summary after literature review.
  2. 05 ÷ 0.8 mm (prepared before the test), Clarify the symbol ÷ in table 1.
  3. Provide the turning test experimental set up including SEM) ZEISS Auriga Compact, Hommeltester, dynamometer Kistler
  4. On what basis feed, cutting depth and cutting speed has been chosen.
  5. It can be seen that the more developed crater wear increases the distance between

the neighboring segments, their size, and degree of chip segmentation, as well as the corresponding segmentation frequency as a result of prolonged accumulation of stress ahead of the cutting edge and its abrupt release when the segment is initiated by the brittle cracking on the free surface. Such behaviour has been already reported [20].

Break this sentence and add more references.

  1. Write a few more future recommendations in the conclusion section.

Author Response

Reviewer: Add literature summary after literature review.

Response: we added some references and introduced the novelty better

Manuscript:

Ramesh et al. [12] also investigated microstructure of WL of hardened steel 52100 and discussed the significance of thermal cycle. The deep insight into thermal cycle during formation of WL was reported by Hosseini et al. [13] as well. It should be mentioned that surface state of quenched steel remarkably affects its functional properties as it was reported in the new studies [14, 15, 16]. Contribution of WL depends on the manner in which surface after hard turning is exploited [17, 18]. Presence of WL is acceptable in the sliding contact whereas the cycling rolling contact is quite sensitive to the surface re-hardening [17 - 19].     

As contrasted with the previous studies [4, 5, 8, 13], this study provides deeper insight into pure contribution of the shear and normal components in the different regions and their relationship to the thermal load. As compared with the study of Wang and Liu [7] or our previous work [6], this study also discusses contribution of the thermal cycle duration as well as the role of specific heat to the thickness of WL. Finally, as compared with [6] thermal cycle and the produced WL are investigated as a function of cutting speed and supplemented by the measurement of microhardness in WL.  

Reviewer: 05 ÷ 0.8 mm (prepared before the test), Clarify the symbol ÷ in table 1.

Response: we replaced symbol ÷ by wording “from … to”

Manuscript: please check appearance of Table 1.  

Reviewer: Provide the turning test experimental set up including SEM) ZEISS Auriga Compact, Hommeltester, dynamometer Kistler

Response: All details about experiments are listed in chapter 2. From our point of view, the photos of devices are quite idle and provide no further valuable information. Some further details can be found in our previous study.

Neslušan, M.; Uríček, J.; Mičietová, A.; Minárik, P.; Píška, M.; Čilliková, M. Decomposition of cutting forces with respect to chip segmentation and white layer thickness when hard turning 100Cr6. J. Manuf. Proc. 50 2020, 475-484, doi: 10.1016/j.jmapro.2020.01.004.

Manuscript: On the other hand, we added some comments and further infotmation about measurement forces, surface roughness as well as microhardness.

Fc as well as Fp were obtained by averaging the three repetitive passes after the signal filtration (the low pass filter 20 Hz).

Five repetitive measurements of WL thickness, microhardness as well as surface roughness were carried out (average values as well as the corresponding standard deviations were calculated and provided in the paper).

Microhardness measurements were carried out using an Innova Test 400TM (50 g for 10 s) on the cross sectional cuts after metallographic observations.

Reviewer: On what basis feed, cutting depth and cutting speed has been chosen.

Response: Cutting feed, depth and speed were chosen on the base of recommendations provided by the producer of inserts. Moreover, these cutting conditions can be considered as the conventional parameters for hard machining cycles when especially the cutting depth and feed should be kept low with respect of thermal softening effect in the cutting zone.

Manuscript: we added this comment.

Cutting feed, depth and speed were chosen on the base of recommendations provided by the producer of inserts. Moreover, these cutting conditions can be considered as the conventional parameters for hard machining cycles when especially the cutting depth and feed should be kept low with respect of thermal softening effect in the cutting zone.

Reviewer: It can be seen that the more developed crater wear increases the distance between

the neighboring segments, their size, and degree of chip segmentation, as well as the corresponding segmentation frequency as a result of prolonged accumulation of stress ahead of the cutting edge and its abrupt release when the segment is initiated by the brittle cracking on the free surface. Such behaviour has been already reported [20].

Break this sentence and add more references.

Response: we added the required referring

Manuscript:

It can be seen that the more developed crater wear increases the distance between the neighbouring segments, their size, and degree of chip segmentation [6], as well as the corresponding segmentation frequency [22]. This is a result of prolonged accumulation of stress ahead of the cutting edge and its abrupt release when the segment is initiated by the brittle cracking on the free surface [26].

Reviewer: Write a few more future recommendations in the conclusion section.

Response: We altered conclusions

Manuscript:

It can be concluded that the influence of cutting edge wear on hard turning is complex. The crater wear alters the shear instability in the tool-chip interface and the corresponding chip appearance. On the other hand, Fat  and Fatn grow with VB and contribute to the thicker WL. However, WL is a product of energy consumed in the tool flank region and the superimposing thermal cycles. The time within the machined surface is exposed to heating plays significant role together with the specific heat strongly affected by cutting speed.

From the practice point of view, it can be reported that VB should not exceed 0.2 mm. The WL in this region is discontinuous or its thickness is low. Employment of the insert of higher VB results into remarkable increase of WL thickness. Employment of the inserts of higher VB could be allowed for roughing cycles only since the thick WL will be removed during the consecutive grinding or turning finishing. Intensification of hard turning cycles via higher cutting speed increases WL thickness but drops down its microhardness.    

Reviewer 2 Report

In this version of the article, the authors show results of a surface after turning steel with medium hardness. In addition, the authors have characterized its white layer (WL) thickness and microhardness, surface roughness as a function of VB tool flank, wear and cutting speed.

The work focuses on analyzing the mechanical and thermal loading of the machined surface to provide a deeper insight into their superimposed contribution. The result is interesting, but it does not contain an innovative element or deep interest for the community. On the other hand, the contributions on how the shear energy is expressed in terms of shear force and how this mechanical energy expressed in terms of the shear components of F grows with VB is converted into heat and strongly affects the thickness of the hardened layer is widely known and expected.

Likewise, the manuscript is susceptible to improvements. For example, minor issues such as:

Scales have been lost in Figure 5. SEM.
What is the error propagation in the measurements of Figure 4? Evolution of Fc and Fp?

These are just some examples of aspects that can be improved.

It is highly recommended that the authors address newer applications for these steels; perhaps tribological studies based on newly implanted ions, analysis in corrosive media, or even uses in additive manufacturing media are the current trend for these steels. Of course, the work is pertinent and the results valuable, but a more integrative discussion is necessary with the current data.

  • Erişir, E., Ararat, Ö., & Bilir, O. G. (2022). Enhancing Wear Resistance of 100Cr6 Bearing Steels by New Heat Treatment Method. Metallurgical and Materials Transactions A, 1-11.
  • Khayatzadeh, A., Sippel, J., Guth, S., Lang, K. H., & Kerscher, E. (2022). Influence of a Thermo-Mechanical Treatment on the Fatigue Lifetime and Crack Initiation Behavior of a Quenched and Tempered Steel. Metals12(2), 204.
  • Reichelt, M., & Cappella, B. (2022). Wear Volume of Self-Mated Steel at the Submicron-Scale: An Atomic Force Microscopy Study. Journal of Tribology144(6).

Author Response

Reviewer: Scales have been lost in Figure 5. SEM.

Response: Sorry for this mistake. We corrected this item.

Manuscript: please check the appearance of Fig. 5.

Reviewer: What is the error propagation in the measurements of Figure 4? Evolution of Fc and Fp?

Response: Standard deviations for calculated components illustrated in Fig. 4 can be recalculated from those for measured Fc and Fp.

Manuscript: We added Nnte into the heading of Fig. 4.

Note: the standard deviations of obtained Fγc is ± 7 N and ± 13 N for Fγp.

Reviewer: It is highly recommended that the authors address newer applications for these steels; perhaps tribological studies based on newly implanted ions, analysis in corrosive media, or even uses in additive manufacturing media are the current trend for these steels. Of course, the work is pertinent and the results valuable, but a more integrative discussion is necessary with the current data.

  • Erişir, E., Ararat, Ö., & Bilir, O. G. (2022). Enhancing Wear Resistance of 100Cr6 Bearing Steels by New Heat Treatment Method. Metallurgical and Materials Transactions A, 1-11.
  • Khayatzadeh, A., Sippel, J., Guth, S., Lang, K. H., & Kerscher, E. (2022). Influence of a Thermo-Mechanical Treatment on the Fatigue Lifetime and Crack Initiation Behavior of a Quenched and Tempered Steel. Metals, 12(2), 204.
  • Reichelt, M., & Cappella, B. (2022). Wear Volume of Self-Mated Steel at the Submicron-Scale: An Atomic Force Microscopy Study. Journal of Tribology, 144(6).

Response: we added these papers into the list of references and introduced their significance in the Introduction.

Manuscript: please check the introduction as well as the list of references.

It should be mentioned that surface state of quenched steel remarkably affects its functional properties as it was reported in the new studies [14, 15, 16]. Contribution of WL depends on the manner in which surface after hard turning is exploited [17, 18]. Presence of WL is acceptable in the sliding contact whereas the cycling rolling contact is quite sensitive to the surface re-hardening [17 - 19].     

  1. Erişir, E.; Ararat, Ö.; Bilir, O. G. Enhancing Wear Resistance of 100Cr6 Bearing Steels by New Heat Treatment Method. Metall. Mater. Trans. A 2022, 53, 1-11. doi: 10.1007/s11661-021-06556-3.
  2. Khayatzadeh, A.; Sippel, J.; Guth, S.; Lang, K. H.; Kerscher, E. Influence of a Thermo-Mechanical Treatment on the Fatigue Lifetime and Crack Initiation Behavior of a Quenched and Tempered Steel. Met. 2022, 12, 204. doi:10.3390/met12020204.
  3. Reichelt, M.; Cappella, B. Wear Volume of Self-Mated Steel at the Submicron-Scale: An Atomic Force Microscopy Study. J. Tribol. 2022, 144, 061702. doi: 10.1115/1.4052963.

Reviewer 3 Report

In this paper, the authors presented the results of investigating the white layer of high tempered bearing steel machined surface related to tool flank wear and cutting speed.  

This is the topic widely investigated and it is well known that the tool flank wear and cutting speed have impact on the properties of the white layer (i.e. generated thermal and mechanical loads). Therefore, the authors have to highlight and prove the novelty in their research. Besides, they must describe the negative influence of the white layer on the performance of the bearings. Furthermore, since the authors have varied different depth of cuts, VBs and cutting speeds, they might determine the optimal ones for which the properties of WL will be acceptable, or for which the avoidance of the white layer can happened.

Why the cooling/lubricating was not mentioned?  

The authors are encouraged to take into the consideration the following comments and suggestions as well: 

Of the 20 references, the authors cited their six references (30%). That is too much.

„The mechanical and thermal load of the machined surface is analysed…“ -  „The mechanical and thermal load of the machined surface were analysed…“

„Cutting energy expressed in term of cutting force is analyses…“ – „Cutting energy expressed in term of cutting force was analysed…“

 „…affects the thickness of the re-hardened layer…“ - „…affects the thickness of the re-hardened layer, WL…“

It would be a much more appropriate to cite some book or theoretical literature instead of research articles 1 and 2, in line 31, because the related sentence (lines 30, 31) is a theoretical, i. e. general statement. The same is valid for [2,3] – line 38. The best option, actually, is to remove the above-mentioned references from these places.

In Figure 1, the marks and the word Tool are red underlined as errors.

Please, compare the following two sentences regarding the cited literature: “WL thickness is affected mainly by the size of VB region and the main cutting motion vc [5-7].” and “Because the WL thickness is predominatly a function of VB [2, 4, 5], the inserts with sufficiently high VB were used in this study to make WL as thick as possible.”

“…alterations was…” – please, check this regarding the English language.

Table 1 – “Inset geometry” – letter “r” is missing.

It would be better to replace the related text, above Figure 3 and Table 2. The same is valid for Figure 4, Figure 8, Figure 9, Figure 10, Figure 12.

It seems that Figure 2 and Figure 3 does not match; or some average values are represented in Figure 3, or…?

 “The evolution of VB versus ap appears to be linear…” - “The evolution of Fc and Fp versus ap appears to be linear…”?

Six SEM images in Figure 5 – the right part of each image is missing.

It would be good to end the section 3 with some text, rather than figure.

Section 3 can be renamed to Results and discussion.

Conclusion is too much summarized.

Author Response

Reviewer: Therefore, the authors have to highlight and prove the novelty in their research. Besides, they must describe the negative influence of the white layer on the performance of the bearings. Furthermore, since the authors have varied different depth of cuts, VBs and cutting speeds, they might determine the optimal ones for which the properties of WL will be acceptable, or for which the avoidance of the white layer can happened.

Response: We added explanation with respect of novelty, negative influence of WL and discuss the stud from the practice point of view.

Manuscript:

It should be mentioned that surface state of quenched steel remarkably affects its functional properties as it was reported in the new studies [14, 15, 16]. Contribution of WL depends on the manner in which surface after hard turning is exploited [17, 18]. Presence of WL is acceptable in the sliding contact whereas the cycling rolling contact is quite sensitive to the surface re-hardening [17 - 19].     

As contrasted with the previous studies [4, 5, 8, 13], this study provides deeper insight into pure contribution of the shear and normal components in the different regions and their relationship to the thermal load. As compared with the study of Wang and Liu [7] or our previous work [6], this study also discusses contribution of the thermal cycle duration as well as the role of specific heat to the thickness of WL. Finally, as compared with [6] thermal cycle and the produced WL are investigated as a function of cutting speed and supplemented by the measurement of microhardness in WL.  

From the practice point of view, it can be reported that VB should not exceed 0.2 mm. The WL in this region is discontinuous or its thickness is low. Employment of the insert of higher VB results into remarkable increase of WL thickness. Employment of the inserts of higher VB would be allowed in roughing cycles only since the thick WL will be removed during the consecutive grinding or turning finishing. Intensification of hard turning cycles via higher cutting speed increases WL thickness but drops down its microhardness.   

Reviewer: Why the cooling/lubricating was not mentioned?

Response: Experiments were carried out in dry conditions. Plastic deformation during hard turning is usually due to remarkable thermal softening. For this reasons, we preferred dry machining.

Manuscript: we added note in tab. 1 that turning was carried out in dry conditions. 

Reviewer: Of the 20 references, the authors cited their six references (30%). That is too much.

Response: We added some new references and replaced some.

Manuscript: Please check the list of references.

Reviewer: „The mechanical and thermal load of the machined surface is analysed...“ - „The mechanical and thermal load of the machined surface were analysed...“

Response: corrected

Manuscript: please check the manuscript

Reviewer: „Cutting energy expressed in term of cutting force is analyses...“ – „Cutting energy expressed in term of cutting force was analysed...“

Response: corrected

Manuscript: please check the manuscript

Reviewer: It would be a much more appropriate to cite some book or theoretical literature instead of research articles 1 and 2, in line 31, because the related sentence (lines 30, 31) is a theoretical, i. e. general statement. The same is valid for [2,3] – line 38. The best option, actually, is to remove the above mentioned references from these places.

Response: We altered referring.

Manuscript: Please check it in the manuscript as well as in the list of refernces.

Reviewer: In Figure 1, the marks and the word Tool are red underlined as errors.

Response: corrected

Manuscript: please check the appearance of Fig. 1.

Reviewer: Please, compare the following two sentences regarding the cited literature: “WL thickness is affected mainly by the size of VB region and the main cutting motion vc [5-7].” and “Because the WL thickness is predominatly a function of VB [2, 4, 5], the inserts with sufficiently high VB were used in this study to make WL as thick as possible.”

Response: We agree with reviewer that the second sentence should be altered.

Manuscript: First sentence was kept in its original version. Second sentence was replaced by the following text.

The inserts with sufficiently high VB were used in this study to make WL as thick as possible. Furthermore, the influence of variable cutting speed (as the additional aspects remarkably contributing to WL thickness) was investigated as well.

Reviewer: “...alterations was...” – please, check this regarding the English language.

Response: corrected …were…”

Manuscript: please check the manuscript

Reviewer: Table 1 – “Inset geometry” – letter “r” is missing

Response: added

Manuscript: please check table 1

Reviewer: It would be better to replace the related text, above Figure 3 and Table 2. The same is valid for Figure 4, Figure 8, Figure 9, Figure 10, Figure 12.

Response: We have made the required text reorganisation.

Manuscript: Please check the manuscript.

Reviewer: It seems that Figure 2 and Figure 3 does not match; or some average values are represented in Figure 3, or...?

“The evolution of VB versus ap appears to be linear...” - “The evolution of Fc and Fp versus ap appears to be linear...”?

Response: To plot Fig. 3 we used the average values of Fc and Fp obtained from the repetitive passes. Records in Fig. 2 represent the example of one pass. We also added explanation associated with the linear fit.  

Manuscript:

It is known that the cutting forces grow along with cutting depth is driven by the power law but the exponent for the cutting depth is near by 1 [3]. For this reason, the linear fit for the measured components is reasonable.

Reviewer: Six SEM images in Figure 5 – the right part of each image is missing.

Response: corrected

Manuscript: please check the appearance of this figure

Reviewer: It would be good to end the section 3 with some text, rather than figure.

Response: corrected

Manuscript: please check the end of section 3.

Reviewer: Section 3 can be renamed to Results and discussion.

Response: renamed

Manuscript: new title of the section as required

Reviewer: Conclusion is too much summarized.

Response: We altered the conclusions.

Manuscript:

It can be concluded that the influence of cutting edge wear on hard turning is complex. The crater wear alters the shear instability in the tool-chip interface and the corresponding chip appearance. On the other hand, Fat  and Fatn grow with VB and contribute to the thicker WL. However, WL is a product of energy consumed in the tool flank region and the superimposing thermal cycles. The time within the machined surface is exposed to heating plays significant role together with the specific heat strongly affected by cutting speed.

From the practice point of view, it can be reported that VB should not exceed 0.2 mm. The WL in this region is discontinuous or its thickness is low. Employment of the insert of higher VB results into remarkable increase of WL thickness. Employment of the inserts of higher VB could be allowed for roughing cycles only since the thick WL will be removed during the consecutive grinding or turning finishing. Intensification of hard turning cycles via higher cutting speed increases WL thickness but drops down its microhardness.    

Reviewer 4 Report

I suggest some points for improvement of the paper as indicated below.

  1. The Introduction section should be supplemented with references. This will give the authors a better understanding of the importance of their research.
  2. Currently innovation and scientific contribution hidden. The last paragraph of the Introduction section highlights innovation and scientific contribution.
  3. Reference number 6 is cited in many places. How different is this research in relation to: Neslušan, M.; Uríček, J.; Mičietová, A.; Minárik, P.; Píška, M.; Čilliková, M. Decomposition of cutting forces with respect to chip segmentation and white layer thickness when hard turning 100Cr6. J. Manuf. Proc. 50 2020, 475-484, doi: 10.1016/j.jmapro.2020.01.004.
  4. Elaborate in detail the selection of the turning process conditions shown in the article and in Table 1.
  5. Is it possible to vary the corner radius?
  6. Measurement errors are not listed in the context of the data shown in Table 2. The impact of measurement errors has not been analyzed. Complete the article with a few sentences.
  7. Equations 1, 2 and 3 must be elaborated in detail in the article. How they were obtained? Elaborate in detail. Check the units in the equations.
  8. Whether the results obtained can be statistically processed?
  9. Can simulation and / or optimization be performed?
  10. Figure 6 shows metallographic images of machined surfaces for a utting speed of 100m / min. Wouldn't it be better to show morphological images for two limit cutting speeds (minimum and maximum), ie. for 70 m / min and 230 m / min.
  11. Discuss the practical application of your research. How it can be applied in industry.
  12. The Conclusions section needs to be supplemented. Highlight the innovation of your methodology. Every research has some shortcomings. List the limitations of your methodology.

Author Response

Reviewer: The Introduction section should be supplemented with references. This will give the authors a better understanding of the importance of their research.

Currently innovation and scientific contribution hidden. The last paragraph of the Introduction section highlights innovation and scientific contribution.

Response: We added some text.

Manuscript:

Ramesh et al. [12] also investigated microstructure of WL of hardened steel 52100 and discussed the significance of thermal cycle. The deep insight into thermal cycle during formation of WL was reported by Hosseini et al. [13] as well. It should be mentioned that surface state of quenched steel remarkably affects its functional properties as it was reported in the new studies [14, 15, 16]. Contribution of WL depends on the manner in which surface after hard turning is exploited [17, 18]. Presence of WL is acceptable in the sliding contact whereas the cycling rolling contact is quite sensitive to the surface re-hardening [17 - 19].     

As contrasted with the previous studies [4, 5, 8, 13], this study provides deeper insight into pure contribution of the shear and normal components in the different regions and their relationship to the thermal load. As compared with the study of Wang and Liu [7] or our previous work [6], this study also discusses contribution of the thermal cycle duration as well as the role of specific heat to the thickness of WL. Finally, as compared with [6] thermal cycle and the produced WL are investigated as a function of cutting speed and supplemented by the measurement of microhardness in WL.  

Reviewer: Reference number 6 is cited in many places. How different is this research in relation to: Neslušan, M.; Uríček, J.; Mičietová, A.; Minárik, P.; Píška, M.; Čilliková, M. Decomposition of cutting forces with respect to chip segmentation and white layer thickness when hard turning 100Cr6. J. Manuf. Proc. 50 2020, 475-484, doi: 10.1016/j.jmapro.2020.01.004.

Response: we explained the novelty as compared with our previous study.

Manuscript:

As compared with the study of Wang and Liu [7] or our previous work [6], this study also discusses contribution of the thermal cycle duration as well as the role of specific heat to the thickness of WL. Finally, as compared with [6] thermal cycle and the produced WL are investigated as a function of cutting speed and supplemented by the measurement of microhardness in WL.  

Reviewer: Elaborate in detail the selection of the turning process conditions shown in the article and in Table 1.

Response: Cutting conditions were selected on the base of the inserts recommendations and with respect of specific character of hard machining. 

Manuscript:

Cutting feed, depth and speed were chosen on the base of recommendations provided by the producer of inserts. Moreover, these cutting conditions can be considered as the conventional parameters for hard machining cycles when especially the cutting depth and feed should be kept low with respect of thermal softening effect in the cutting zone.

Reviewer: Is it possible to vary the corner radius?

Response: Yes, it is. Corner radius can be altered during manufacturing process – rectification of cutting edge. However, in our particular case, corner radius is altered as a result of cutting edge wear.  

Manuscript: We prefer no change.

Reviewer: Measurement errors are not listed in the context of the data shown in Table 2. The impact of measurement errors has not been analyzed. Complete the article with a few sentences.

Response: We agree with reviewer. Table provides standard reviations for VB, edge radius as well as rake angle. Standard deviations for calculated components indicated in Tab. 2 can be obtained from those for measured Fc and Fp.

Manuscript: We added note.

Note: the standard deviations of obtained Fαt is about ± 7 N and ± 13 N for Fαtn.

Reviewer: Equations 1, 2 and 3 must be elaborated in detail in the article. How they were obtained? Elaborate in detail. Check the units in the equations.

Response: These equations (especially 1 and 2) are very well known for many years. For this reason, we prefer no change in the manuscript. Origin of equation 3 is explained in the text.

Manuscript: We prefer no change.

Reviewer: Whether the results obtained can be statistically processed?

Response: We carried out five repetitive measurements for each experimental measurement such as components of cutting force, WL thickness, WL hardness as well as surface roughness. Moreover, we calculated correlation coefficients in Fig. 12, Fig. 14 (also P value in Fig. 14). Average values and standard deviations were also calculated from the repetitive measurements. Extensive statistical processing would require more data. However, we consider that the provided data are reliable.

Manuscript: We added this text.

Fc as well as Fp were obtained by averaging the three repetitive passes after the signal filtration (the low pass filter 20 Hz).

Five repetitive measurements of WL thickness, microhardness as well as surface roughness were carried out (average values as well as the corresponding standard deviations were calculated and provided in the paper).

Reviewer: Can simulation and / or optimization be performed?

Response: It is quite out of scope of this manuscript. However, we are sceptic that simulation can be carried out in the reliable manner due to very complicated state in the cutting zone – very strong process dynamics, very high temperatures, phase transformation, ect.

With respect of optimisation we did not alter cutting conditions and only investigate influence of VB. It can be found that VB should not exceed 0.2 mm because of remarkable increase of white layer thickness.

Manuscript: We added comment into conclusions. From the practice point of view, it can be reported that VB should not exceed 0.2 mm. The WL in this region is discontinuous or its thickness is low. Employment of the insert of higher VB results into remarkable increase of WL thickness.

Reviewer: Figure 6 shows metallographic images of machined surfaces for a utting speed of 100m / min. Wouldn't it be better to show morphological images for two limit cutting speeds (minimum and maximum), ie. for 70 m / min and 230 m / min.

Response: We prefer to keep appearance of this figure as it is in the original version of manuscript.

Manuscript: We prefer no change.

Reviewer: The Conclusions section needs to be supplemented. Highlight the innovation of your methodology. Every research has some shortcomings. List the limitations of your methodology. Discuss the practical application of your research. How it can be applied in industry.

Response: We altered the conclusions

Manuscript:

It can be concluded that the influence of cutting edge wear on hard turning is complex. The crater wear alters the shear instability in the tool-chip interface and the corresponding chip appearance. On the other hand, Fat  and Fatn grow with VB and contribute to the thicker WL. However, WL is a product of energy consumed in the tool flank region and the superimposing thermal cycles. The time within the machined surface is exposed to heating plays significant role together with the specific heat strongly affected by cutting speed.

From the practice point of view, it can be reported that VB should not exceed 0.2 mm. The WL in this region is discontinuous or its thickness is low. Employment of the insert of higher VB results into remarkable increase of WL thickness. Employment of the inserts of higher VB could be allowed for roughing cycles only since the thick WL will be removed during the consecutive grinding or turning finishing. Intensification of hard turning cycles via higher cutting speed increases WL thickness but drops down its microhardness.    

Reviewer 5 Report

The paper presents a well-developed experimental and microstructure investigation on surface state analysis after the turning process. 

Please correct the manuscript based on the following comments:

Please explain the paper novelty in more interesting manner. 

Abstract and introduction parts should be improved. Introduction section is noticed to have limited literature survey about the research topic. There should be proper continuation on literature survey and how the research gap exists on this topic. Similarly, why this research topic is important?. 

Also in Figures 3(a) and 3(b), the data points are fitted with linear regression model. Give more information about the fitted lines in terms of statistical point of view.  

in Figure 5, magnification scales are not visible. It seems they are cut down from the graphs. Please add it by moving them little inside the graphs.

 In Figures 6(a)-(f), increase the magnifications scale font size.   

Conclusions part looks boring. Please make it more interesting with some notable outcomes.

Thank You.  

Author Response

Reviewer: Please explain the paper novelty in more interesting manner.

Abstract and introduction parts should be improved. Introduction section is noticed to have limited literature survey about the research topic. There should be proper continuation on literature survey and how the research gap exists on this topic. Similarly, why this research topic is important?.

Response: we added some text into Introduction.

Manuscript:

Ramesh et al. [12] also investigated microstructure of WL of hardened steel 52100 and discussed the significance of thermal cycle. The deep insight into thermal cycle during formation of WL was reported by Hosseini et al. [13] as well. It should be mentioned that surface state of quenched steel remarkably affects its functional properties as it was reported in the new studies [14, 15, 16]. Contribution of WL depends on the manner in which surface after hard turning is exploited [17, 18]. Presence of WL is acceptable in the sliding contact whereas the cycling rolling contact is quite sensitive to the surface re-hardening [17 - 19].     

As contrasted with the previous studies [4, 5, 8, 13], this study provides deeper insight into pure contribution of the shear and normal components in the different regions and their relationship to the thermal load. As compared with the study of Wang and Liu [7] or our previous work [6], this study also discusses contribution of the thermal cycle duration as well as the role of specific heat to the thickness of WL. Finally, as compared with [6] thermal cycle and the produced WL are investigated as a function of cutting speed and supplemented by the measurement of microhardness in WL.  

Reviewer: Also in Figures 3(a) and 3(b), the data points are fitted with linear regression model. Give more information about the fitted lines in terms of statistical point of view. 

Response: We added explanation.

Manuscript: we added text as well as reference.

It is known that the cutting forces grow along with cutting depth is driven by the power law but the exponent for the cutting depth is near by 1 [21]. For this reason, the linear fit for the measured components is reasonable.

  1. Neslušan, M. Turning of hardened steels. first ed., University of Žilina, 2010.

Reviewer: in Figure 5, magnification scales are not visible. It seems they are cut down from the graphs. Please add it by moving them little inside the graphs.

Response: Sorry for this mistake. We corrected this item.

Manuscript: please check the appearance of Fig. 5.

Reviewer: In Figures 6(a)-(f), increase the magnifications scale font size.  

Response: we have mad et he images in Fig. 6 larger.

Manuscript: please check the appearance of Fig. 6.

Reviewer: Conclusions part looks boring. Please make it more interesting with some notable outcomes.

Response: We altered the conclusions.

Manuscript:

It can be concluded that the influence of cutting edge wear on hard turning is complex. The crater wear alters the shear instability in the tool-chip interface and the corresponding chip appearance. On the other hand, Fat  and Fatn grow with VB and contribute to the thicker WL. However, WL is a product of energy consumed in the tool flank region and the superimposing thermal cycles. The time within the machined surface is exposed to heating plays significant role together with the specific heat strongly affected by cutting speed.

From the practice point of view, it can be reported that VB should not exceed 0.2 mm. The WL in this region is discontinuous or its thickness is low. Employment of the insert of higher VB results into remarkable increase of WL thickness. Employment of the inserts of higher VB could be allowed for roughing cycles only since the thick WL will be removed during the consecutive grinding or turning finishing. Intensification of hard turning cycles via higher cutting speed increases WL thickness but drops down its microhardness.    

Round 2

Reviewer 2 Report

The authors have incorporated in this new version most of the changes. 

Reviewer 3 Report

The authors have taken into the consideration all the comments and suggestions. The work is significantly improved.

Reviewer 4 Report

The authors responded to comments and revised the manuscript. The manuscript has been improved.